

# Be aware of the allele-specific bias and compositional effects in multi-template PCR

Ilia Korvigo[1,2,*,†], Anna A. Igolkina[2,3,*], Arina A. Kichko[2], Tatiana Aksenova[2] and Evgeny E. Andronov[2,4]

[1] Faculty of Infocommunication Technologies, ITMO University, St. Petersburg, Russia
[2] Laboratory of Microbiological Monitoring and Bioremediation of Soils, All-Russia Research Institute for Agricultural Microbiology, St. Petersburg, Russia
[3] GMI—Gregor Mendel Institute of Molecular Plant Biology, Vienna, Austria
[4] Dokuchaev Soil Science Institute, Moscow, Russia
[*] These authors contributed equally to this work.
[†] Deceased.

Corresponding authors
Ilia Korvigo, eeandr@gmail.com, ilia.korvigo@gmail.com
Anna A. Igolkina, igolkinaanna11@gmail.com

## ABSTRACT

High-throughput sequencing of amplicon libraries is the most widespread and one of the most effective ways to study the taxonomic structure of microbial communities, even despite growing accessibility of whole metagenome sequencing. Due to the targeted amplification, the method provides unparalleled resolution of communities, but at the same time perturbs initial community structure thereby reducing data robustness and compromising downstream analyses. Experimental research of the perturbations is largely limited to comparative studies on different PCR protocols without considering other sources of experimental variation related to characteristics of the initial microbial composition itself. Here we analyse these sources and demonstrate how dramatically they effect the relative abundances of taxa during the PCR cycles. We developed the mathematical model of the PCR amplification assuming the heterogeneity of amplification efficiencies and considering the compositional nature of data. We designed the experiment—five consecutive amplicon cycles (22–26) with 12 replicates for one real human stool microbial sample—and estimated the dynamics of the microbial community in line with the model. We found the high heterogeneity in amplicon efficiencies of taxa that leads to the non-linear and substantial (up to fivefold) changes in relative abundances during PCR. The analysis of possible sources of heterogeneity revealed the significant association between amplicon efficiencies and the energy of secondary structures of the DNA templates. The result of our work highlights non-trivial changes in the dynamics of real-life microbial communities due to their compositional nature. Obtained effects are specific not only for amplicon libraries, but also for any studies of metagenome dynamics.

## INTRODUCTION

High-throughput sequencing (HTS) of marker gene amplicon libraries is by far the most popular and accessible way to study microbial communities regardless of the environment they inhabit. Thanks to targeted amplification, it provides unparalleled sensitivity and competitive functional reconstructions at lower costs than shotgun metagenomic sequencing (*Langille et al., 2013*), especially in low-entropy microbial communities. At the same time, amplification is an imperfect process that adds a whole layer of noise and variation detrimental to downstream analyses. PCR artefacts, such as erroneous copies and chimera, are perhaps the most notorious of these side-effects and as such have received a lot of attention in the literature and prompted the development of effective error models and denoising algorithms (*Quince et al., 2011*; *Rosen et al., 2012*; *Callahan et al., 2016*; *Amir et al., 2017*) that are now widely employed in microbiome analysis workflows.

Another source of perturbations in the initial taxonomic structure is selective amplification which potentially leads to unreliable and spurious statistical results in microbiome research based on 16S amplicon data. That is not to say that these perturbations are a totally obscure and arbitrary phenomenon; however, it was not studies deeply enough. A recent review of opinions on experimental biases in amplicon sequencing data (*Eisenstein, 2018*) shows that workflow standardisation is considered sufficient to mitigate statistical artifacts associated with PCR (as well as other similar selective procedures). However, this approach is based on an as yet unverified premise that similar experimental conditions (*i.e.,* PCR protocols) lead to reproducible perturbations across diverse communities. While this line of reasoning might not be misguided in general (though there is growing evidence to that end (*Jones et al., 2015*)), in our particular case it goes against a whole body of published multi-template PCR simulations and molecular studies on quantitative single-template PCR.

We can highlight four major molecular effects contributing to amplification disparity: (i) differences in primer-template binding energies (a function of the primer binding-site's sequence), (ii) self-annealing probability (discriminating against highly abundant templates) (*Paliy & Foy, 2011*; *Chatterjee, Banerjee & Datta, 2012*; *Kalle, Kubista & Rensing, 2014*; *Kebschull & Zador, 2015*; *Peng et al., 2018*), (i) unequal denaturation of templates as a function of GC content (iv) and stability of secondary structures (*Fan et al., 2019*). Moreover, an important effect was reported by *Gonzalez et al. (2012)* where it was shown that the relative amplification efficiency for each bacterial species is a nonlinear function of the fraction that each of those taxa represent within a community or multispecies DNA template. Consequently, the low-proportion taxa in a community are under-represented during PCR-based surveys. In addition to that, the first few PCR cycles are also subject to purely stochastic variations manifesting as the so-called PCR drift. At least one of these effects, namely self-annealing probability, cannot be invariant to initial community composition, though it is unclear how prominent the effect is.

Experimental microbiome research in this area is largely limited to comparative studies of different PCR protocols that offer little insight into how perturbations translate between communities with different initial states under identical PCR conditions. These studies

make it hard to distinguish between biases imparted by amplification and other selective steps in an amplicon sequencing workflow (*Aird et al., 2011*; *Kennedy et al., 2014*; *Jones et al., 2015*; *Krehenwinkel et al., 2017*). That being said, these studies do show that even minor variations in PCR protocols result in significantly different taxonomic profiles. By extension, these perturbations can alter perceived associations between a community's structure and a biological process of interest.

One of the approaches to assess the PCR bias relays on the use of the mock-mixes, *i.e.,* artificially prepared communities with known initial proportions of a moderate number of taxa (*Pinto & Raskin, 2012*; *Krehenwinkel et al., 2017*; *McLaren, Willis & Callahan, 2019*; *Silverman et al., 2021*; *Yeh et al., 2021*). The drawback of this approach is a poor resemblance of simulated samples to real-life extremely diverse microbiome communities. This insufficient diversity does not allow to analyse possible sources of PCR bias, such as heterogeneity in amplification efficiencies and the impact of the complex compositional structure of the microbial community. In this study, we used an alternative approach. Instead of making mock-mixes we utilized one native microbiome (human stool), and traced dynamics of its relative abundances of taxa in the series of consecutive PCR cycles in replicates. Our data covered a very wide taxonomic spectrum, so that both heterogenous nature of alleles (different GC-composition and secondary structures) and complex structure of community were incorporated in the analysis.

Finally, the compositional nature of the microbial communities is usually neglected in discussions of the PCR bias. We suppose that this topic does not receive much attention due to the complexity of statistics required for the compositional data analysis (CoDA). CoDA is widely applied nowadays in geology, biology, economics, and social sciences to analyse and correctly reanalyse compositional data (https://www.coda-association.org). The high-throughput sequencing data are inherently compositional: only relative information between abundances is reasonable (*Gloor et al., 2017*; *Quinn et al., 2018*), so that microbial communities are traditionally described by proportions of its members (*i.e.,* composition). The obvious characteristic of a composition is that proportions of its components are not independent, and the increase of one component in a composition leads to the decrease of all others even the absolute amounts of the latter stay without changes. Therefore, the analysis of proportions requires non-conventional statistical methods under specific compositional assumptions. Here we present the model of PCR amplification under these assumptions, revealed the composition-dependent PCR bias: complex and non-linear changes of relative abundancies during the amplification and use the data obtained for identification of mechanisms responsible for allele-specific bias in multi-template RCR.

## METHODS
### Sample acquisition
#### Ethics statement
This study was approved by the Institutional Review Board of Federal State Budget Scientific Institution "All-Russian Research Institute of Agricultural Microbiology" (FSBSI ARRIAM) #96/06 from 04.06.2020. Informed signed donor consent was obtained prior to participation.
*Sample collection*

One stool specimen was provided by an adult male in January 2018 in St.-Petersburg, Russia, and frozen immediately at $-20\,°C$ until DNA extraction. The donor had no history of enteric illness and experienced no exposure to antibiotics in the past year.

## The calibration experiment

We extracted DNA using a PowerSoil DNA isolation kit (QIAGEN, Venlo, Netherlands) following the manufacturer's protocol. To aid our modelling efforts, we tried to minimise all sources of experimental variance as much as possible. To avoid batch effect, all amplicon libraries were constructed in a single run, effectively limiting the number of libraries to the capacity of a T100 thermocycler (Bio Rad, Hercules, CA, USA) we had at our disposal, *i.e.,* 96. Moreover, being restricted to a single amplification run meant opening the thermocycler's lid and extracting a subset of strip-tubes thereby introducing unwanted disturbances into the amplification process. Therefore, we decided to limit the number of such extraction events and, by extension, the number of cycles to extract. In view of these limitations, we carried out a preliminary qPCR assay mimicking our standard amplicon library preparation procedure to select a range of cycles within the log-linear amplification phase that yielded enough product for subsequent sequencing. The essay were performed with qPCR-HS SYBR (Evrogen, Moscow, Russia) mix accordingly to the manufacturer's recommendation with universal 16S V4 rRNA primers (F515—GTGCCAGCMGCCGCGGTAA and_R806—GGACTACVSGGGTATCTAAT) (*Caporaso et al., 2011*). Accordingly to the qPCR essay cycles 22–26 were selected for the deep amplicon sequencing. Further library preparation procedures followed the Illumina MiSeq Reagent Kit Preparation Guide. A single PCR reaction contained 0.15 µl of high-fidelity Encyclo polymerase (Evrogen, Moscow, Russia), 1.5 µl 10X Encyclo buffer (Evrogen, Moscow, Russia), 0.2 µl of 10 mM dNTP (Evrogen, Moscow, Russia), 5.0 pM of universal 16S V4 rRNA primers, 1.0 µl of extracted DNA (approximately $10^5$ templates) and 11.65 µl of PCR water (Evrogen, Moscow, Russia). To avoid temperature fluctuation associated with the edges of the thermocycler, we decided to only fill a centred eight by 12 grid within the thermocycler with 60 tubes covering cycles 22–26 (12 replicates per cycle). To further minimise biases arising from spatial variations in thermodynamic conditions inside the thermocyler, we randomised the placement of strip-tubes (assuming radial symmetry). The libraries were sequenced on an Illumina MiSeq machine using a MiSeq Reagent Kit v3 (600 cycles). Mean number of reads per sample was 49685 (min: 28104, max: 81,330); after the filtration 19,179 reads per samples were remains (min: 9988, max: 31,171). After the filtering, reads were rarefied to 9988 reads.

## Data preprocessing

We used DADA2 (*Callahan et al., 2016*) to infer exact amplicon sequence variants (ASVs) and remove chimera. As per DADA2 recommendations on error-model inference, ASV reconstruction was carried out in separate PCR-cycle groups. To train an IdTaxa taxonomy classifier (*Murali, Bhargava & Wright, 2018*), we downloaded and preprocessed the SILVA Ref 132 database (*Pruesse et al., 2007*): we extracted the amplified region,

truncated taxonomic descriptions at the highest ambiguous taxonomic rank and removed sequences left without genus-level annotations. To reduce redundancy in the training dataset without mixing different taxa together, we grouped all remaining records by their complete taxonomic identifiers, removed duplicates within the groups using CD-HIT (*Fu et al., 2012*) and subsampled them without replacement to allow at most 10 representatives per group. The classifier was trained for 30 iterations. We predicted taxonomy and used SEPP (*Janssen et al., 2018*) to insert ASVs into the reference GreenGenes 13.8 (99%) phylogenetic tree. Amplicon sequence variants that could not be inserted into the tree or were not identified at the genus level with a confidence level $\geq$ 80% were discarded. To mitigate artifacts of zero-replacement and reduce uncertainty associated with rare amplicon sequence variants, we discarded ASVs that were not observed over 10 times in at least 50% of the libraries. The remaining data were treated by Bayesian-multiplicative zero-replacement with a Dirichlet prior (*Martín-Fernańdez et al., 2015*).

## Isometric log-ratio transform

As frequencies or proportions, HTS data has a compositional nature and requires a particular way of analysis, compositional data analysis (CoDA). A composition is a set of $n$ dependent positive components under the constant sum constraint. To overcome this constraint, a composition should be transformed into the vector of independent components, where the standard analysis for the unconstrained data in real space can be performed. One of these transformations is an isometric log-ratio transform (ILR), which is based on the bipartition strategy developed by *Silverman et al. (2017)* and *Egozcue et al. (2003)*. Briefly, given a rooted binary tree of $n$ leaves (DNA templates) and $n-1$ internal nodes, we can define a sign-matrix $\Phi$ of $n-1$ rows and $n$ columns such that

$$
\Phi_{ij} = \begin{cases} -1, & \text{template } j \text{ belongs to the left subclade of node } i \\ +1, & \text{template } j \text{ belongs to the right subclade of node } i \\ 0, & \text{template } j \text{ does not descend from node } i. \end{cases}
$$

We can now scale rows in matrix $\Phi$ and define contrast matrix $\Psi$ of size $(n-1 \times n)$ such that

$$
\Psi_{ij} = \begin{cases} \Phi_{ij} \dfrac{k_i}{n_{i-}}, & \Phi_{ij} < 0 \\ \Phi_{ij} \dfrac{k_i}{n_{i+}}, & \Phi_{ij} > 0 \\ 0, & \Phi_{ij} = 0 \end{cases}
\tag{1}
$$

where $n_{i+} = \sum_{j=1}^{n}[\Phi_{ij} > 0]$, $n_{i-} = \sum_{j=1}^{n}[\Phi_{ij} < 0]$ and $k_i = \sqrt{\dfrac{n_{i-}n_{i+}}{n_{i-}+n_{i+}}}$. The important property of $\Psi$ matrix is that sums of each its row is equal to zero. Let $x$ be a composition, *i.e.*, vector of $n$ positive components, then its isometric log-ratio transform is

$$
\text{ilr}(x) = \text{ilr}(\mathfrak{C}(x)) = \log(x)\Psi^{T} = b
\tag{2}
$$

where $\mathfrak{C}$ denotes compositional closure (*Pawlowsky-Glahn & Buccianti, 2011*), $b$ is a vector of so-called balances. Since ILR is an isometry on the Aitchison simplex, we can convert

balances back into relative abundances:

$$x = \mathrm{ilr}^{-1}(b) = \mathfrak{C}(\exp(b\Psi)). \tag{3}$$

## PCR amplification model for one sequence

Let $z$ be the amount of an original DNA sequence, $\theta \in (0,1]$ be a constant amplification efficiency associated with the DNA template, and $\lambda \in (0,1]$ be a constant amplification efficiency of its PCR product. Then, the PCR reaction is expressed as the following recurrence relation capturing a continuous generalisation of the number of PCR products, $c$, available at time-step (cycle) $t$:

$$c(t) = \theta z + c(t-1) + \lambda c(t-1) = \theta z + (\lambda+1)c(t-1),$$

where three terms denote amplified product from the original DNA, remained product from the previous cycle and the amplified product. It should be highlighted, that $z$ and $\theta$ variables are present only as a product and are non-identifiable separately. Practically, it means that the initial concentration of a DNA sequence is not detectable in principle, and only an estimate with respect to amplification efficiency, $\theta z$, can be obtained.

$$c(t) - c(t-1) = (\lambda+1)c(t-1) - (\lambda+1)c(t-2)$$
$$\Rightarrow c(t) = (\lambda+2)c(t-1) - (\lambda+1)c(t-2).$$

We now have a second-degree linear homogeneous recurrence relation, where the initial conditions are $c(1) = \theta z$ and $c(2) = (\lambda+1)\theta z + \theta z$. The relation has the following characteristic equation:

$$x^2 - (\lambda+2)x + (\lambda+1) = 0 \Rightarrow \begin{cases} x_1 = (\lambda+1) \\ x_2 = 1. \end{cases}$$

The general solution to this equation is a linear combination of both roots

$$c(t) = \xi(\lambda+1)^t + \zeta 1^t$$

Then, from initial conditions we have

$$\begin{cases} c(1) = \theta z \\ c(2) = (\lambda+1)\theta z + \theta z \end{cases} \Rightarrow \begin{cases} \xi(\lambda+1) + \zeta = \theta z \\ \xi(\lambda+1)^2 + \zeta = (\lambda+1)\theta z + \theta z \end{cases} \Rightarrow \begin{cases} \xi = \dfrac{\theta z}{\lambda} \\ \zeta = -\dfrac{\theta z}{\lambda}. \end{cases}$$

Therefore, under our model the number of PCR products associated with a template at cycle $t$ is given by

$$c(t) = \frac{\theta z}{\lambda}(\lambda+1)^t - \frac{\theta z}{\lambda}. \tag{4}$$

## PCR model for composition data

We model PCR as a discrete-time process parametrised by amplification efficiencies and initial template concentrations, $z = (z_1, \ldots, z_n)$. We distinguish amplification efficiencies associated with original DNA sequences extracted from an environment, $\theta = (\theta_1, \ldots \theta_n)$, and their amplicons, $\lambda = (\lambda_1, \ldots \lambda_n)$, due to stochastic nature of early PCR cycles and differences in template lengths, primer binding-site composition (and, by extension, primer-template complex stability). We assume constant efficiencies (though there is much theoretical and empirical evidence to the contrary (*Paliy & Foy, 2011*; *Chatterjee, Banerjee & Datta, 2012*; *Kalle, Kubista & Rensing, 2014*)) and implicitly restrict our model to the log-linear amplification phase (thereby ignoring potential inter-template competition for substrates at later stages). Under this model (see derivation for Eq. (1)) the number of PCR products associated with template $i$ at cycle $t$ is given by

$$c_i(t) = \frac{\theta_i z_i}{\lambda_i}(\lambda_i + 1)^t - \frac{\theta_i z_i}{\lambda_i} \tag{5}$$

where $z_i > 0$ and $\lambda_i, \theta_i \in (0, 1]$. Assuming $\theta_i = \lambda_i$ and including the initial concentrations of DNA sequences, the Expression 5 becomes equivalent to the traditional monoparametric PCR count approximation:

$$c_i(t) = z_i(\lambda_i + 1)^t. \tag{6}$$

Assuming large values of $t$ (i.e, PCR cycles $>5$), we approximate the Expression 5 as follows:

$$c_i(t) = \frac{\theta_i z_i}{\lambda_i}(\lambda_i + 1)^t - \frac{\theta_i z_i}{\lambda_i} = \frac{\theta_i z_i}{\lambda_i}(\lambda_i + 1)^t\left(1 - \left(\frac{1}{\lambda_i + 1}\right)^t\right) \approx \frac{\theta_i z_i}{\lambda_i}(\lambda_i + 1)^t. \tag{7}$$

While it is impossible to detect absolute amplicon counts, $c_i(t)$, the observed values after the PCR amplification have a compositional nature. Consequently, we use the isometric log-ratio transform (ILR) and model compositions as balances defined on internal nodes of the amplicon phylogenetic tree (see Eqs. (1) and (2)).

Let $c_i'(t)$ be an observed value of $i$th template in a sample after $t$ PCR cycles, then the relation between observed and actual amplicon counts is linear, $c_i'(t) = Ac_i(t)$, where $A$ is sample-specific factor and the same for all templates of the sample. This factor crucially influences the comparison of amplicon counts between samples, and normalization procedures bring an additional source of bias. To overcome the problem with sample-specific factors, ILR transformation (Eq. (2)) is applied to observed values:

$$\mathrm{ilr}\big(c'(t)\big) = \log c'(t)\Psi^T = \log(Ac(t))\Psi^T = \log(c(t))\Psi^T + \big(\log A, \ldots \log A\big)\Psi^T$$
$$= \big[\text{using property of } \Psi\big] = \log(c(t))\Psi^T = \mathrm{ilr}(c(t)).$$

Thus, ILR values of observed amplicon counts equal to ILR values of actual counts. We then substitute Eq. (7) into ILR transformation

$$\mathrm{ilr}(c(t)) = \big(\log(\lambda + 1)t - \log\lambda + \log\theta z\big) \cdot \Psi^T = t\log(\lambda + 1)\Psi^T - \log\lambda \cdot \Psi^T$$
$$+ \log\theta z \cdot \Psi^T, \tag{8}$$

where $\lambda = (\lambda_1, \ldots \lambda_n)$, $\theta z = (\theta_1 z_1, \ldots \theta_n z_n)$.

In the Eq. (8), $\theta z$ is present only once and in the product with $\Psi^T$, so that the term $\log \theta z \cdot \Psi^T$ forms the vector of size $(n-1)$, and $n$ elements of $\theta z$ become unidentifiable. This is not surprising due to the fact, that $\theta z$ reflect biased initial concentrations, which have the compositional nature too. We introduce $a = \log \theta z \cdot \Psi^T = \mathrm{ilr}(\theta z)$, and balances of amplicon counts are treated as follows:

$$b(t) = t \log(\lambda + 1) \Psi^T - \log \lambda \cdot \Psi^T + a. \qquad (9)$$

Eq. (9) demonstrate that the growth of PCA products is linear over $t$ in the space of balances. When all parameters of Eq. (8) are estimated ($\hat{\lambda}$ and $\hat{a}$), one can assess relative amplicon counts at any PCR cycle $t$ as follows:

$$\hat{c}(t) = \mathfrak{C}\left( \frac{\hat{z}_1}{\hat{\lambda}_1}\left( (\hat{\lambda}_1 + 1)^t - 1 \right), \ldots, \frac{\hat{z}_n}{\hat{\lambda}_n}\left( (\hat{\lambda}_n + 1)^t - 1 \right) \right), \qquad (10)$$

where $\hat{z} = \mathrm{ilr}^{-1}(\hat{a})$.

## Bayesian inference

Let $x^{tj} = (x_1^{tj}, \ldots, x_n^{tj})$ be a vector of $n$ measured amplicon counts for $j$th replicate at cycle $t$. Then vector of balances of this observation is $b^{tj} = \mathrm{ilr}(x^{tj})$. The whole set of observations in our study is $\{b^{tj}\}_{t=\overline{22,26}, j=\overline{1,12}}$. We assume that $b^{tj}$ follows the multivariate Gaussian distribution and take a part in the following hierarchical Bayesian model:

$$b^{tj} \sim \mathfrak{N}(b(t), \Sigma),$$
$$b(t) = t \log(\lambda + 1) \cdot \Psi^T - \log \lambda \cdot \Psi^T + a,$$
$$\lambda \sim \mathrm{Beta}(4, 1),$$
$$a \sim \mathfrak{N}(0, 2),$$
$$\Sigma = \mathrm{diag}(\sigma_i^2)$$
$$\sigma_i \sim \mathfrak{N}_+(0, 1).$$

The model was implemented in Python using the probabilistic modelling package PyMC3 (*Salvatier, Wiecki & Fonnesbeck, 2015*). Parameters were inferred using the No-U-Turn Sampler (*Hoffman & Gelman, 2014*), a self-tuning variant of Hamiltonian Monte Carlo, over 5000 tuning samples and 20,000 main samples repeated across 4 MCMC chains. We used leave-one-out cross-validation to optimise parameters of prior distributions. We used the Gelman–Rubin statistic and the effective number of samples to monitor convergence and autocorrelation.

## Energy estimations

Free energies of secondary structures of 16S rRNA sequences were calculated at Mfold server (*Zuker, 2003*; *SantaLucia Jr, 1998*) with the salt correction (*Peyret, 2000*). PCR elongation parameters used: temperature 72 °C, salt 50 mM. From a set of alternative structures, we chose those with maximal values of free energies.

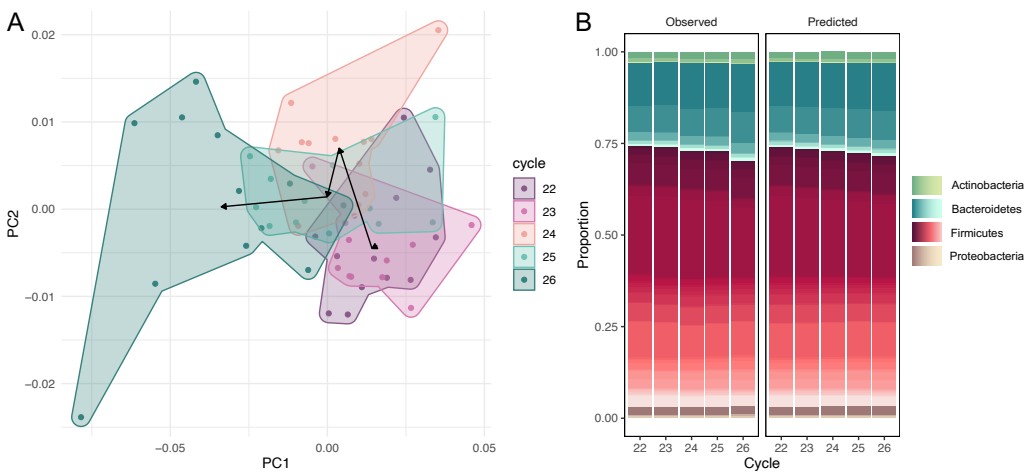

**Figure 1   Dynamic of community over 5 cycles.** (A) Shift of the microbial taxonomic structure with the increase of cycles. (B) A comparison between high-level temporal community dynamics in experimental and inferred data. Inferred dynamics appear to be a smoothed version of observed variations. In both cases we observe a rapid expansion of Bacteroides at the expense of Firmicutes. Relative groupabundances are estimated from 500 samples from the posterior-predictive distribution.

## Data and code availability

Raw sequencing data were deposited in the NCBI Sequence Read Archive (SRA): BioProject PRJNA545409. All code and metadata required to reproduce the study are openly available on GitHub (https://github.com/arriam-lab2/pcr_bias_publication). We used package ggplot2 (*Wickham, 2011*) to generate visualisations.

## RESULTS

### Shift of the microbial community in the PCA amplification

The dataset of amplicon counts consists of 60 observations: five PCR cycles and 12 repeats for each cycle. We performed the amplicon sequence variants picking, and standard PCoA showed the clear shift of the microbial taxonomic structure with the increase of cycles (Fig. 1A). This dynamic is also distinct at the phylum level of the microbial compositions: a rapid expansion of Bacteroides and a much slower growth of Actinobacteria and Proteobacteria at the expense of Firmicutes (Fig. 1B). While, at the resolution of five cycles, these shifts are not very dramatic visually, the PCR bias between the first and 20-th cycles could be significant.

### Comparing observed and predicted community dynamics

High-level community dynamics predicted by the PCR amplification model were visually consistent (albeit smoothed out) with observed variations in experimental data (Fig. 1B). On a finer level, experimental log-ratios between relative abundances of individual amplicon sequence variants measured at cycles 26 and 22 were strongly correlated with inferred log-ratios (Pearson $r = 0.87$, 95% CI = $[0.67, 0.95]$, $p = 0$). These observations suggest that
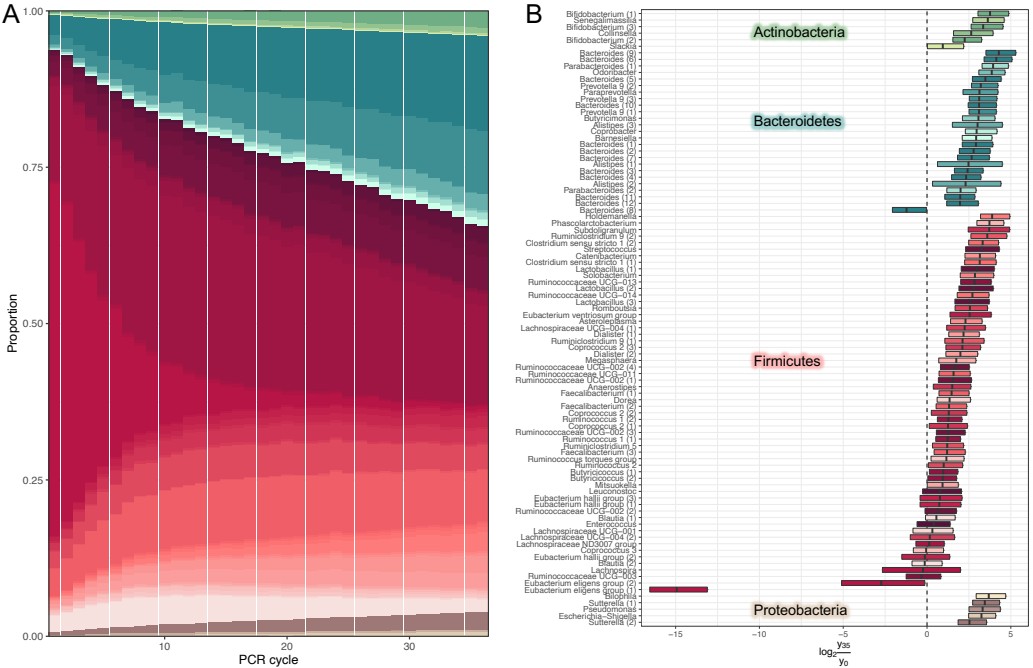

**Figure 2 Estimates of dynamics.** (A) Inferred high-level community dynamics at cycles 0–35. Community composition at cycle 0 corresponds to approximated initial community profile. PCR bias resulted in a dramatic overrepresentation of Bacteroides and a corresponding shrinkage of Firmicutes. Relative group abundances are estimated from 500 samples from the posterior-predictive distribution. (B) Estimated log-ratio distributions between relative abundances of individual amplicon sequence variants (ASVs) inferred at cycles 0 (that is approximated initial template proportions) and 35. Vertical lines inside the boxes represent medians. Boxes represent interquartile distances. Log-ratios were estimated from 500 samples from the posterior-predictive distribution. Most relative abundances changed by a factor of 2–8. The plot contains the subset of amplicon sequence variants classified at the genus level. Numbers in parentheses are used to disambiguate ASVs classified into the same genera.

the amplification model was able to capture and reproduce experimental PCR-induced variations.

## Characterising and mitigating amplification bias

We used our model to predicted large-scale community dynamics from cycle 0 to cycle 35 for all ASVs (amplicon sequence variants). PCR bias leads to a dramatic over-representation of Bacteroides and shrinkage of Firmicutes (Fig. 2A). To better visualize the magnitude of amplification biases introduced by a standard amplicon library preparation workflow, we selected amplicon sequence variants (ASVs) classified at the genus level and estimated log-ratio distributions between their relative abundances inferred at cycles 1 (that is, approximated initial template proportions) and 35 (Fig. 2B). Most relative abundances changed by a factor of 2-8. One particularly unfortunate amplicon sequence variant in the Eubacterium eligens group was affected by a factor of 215. The distribution of log-ratios between two distant cycles makes it tempting to view amplification biases as a linear phenomenon.
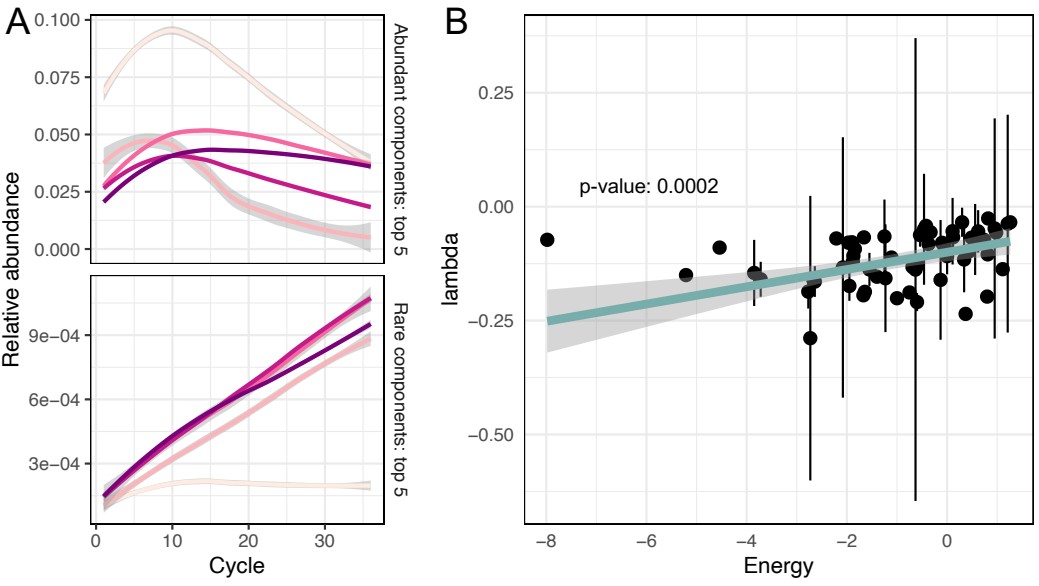

**Figure 3** **Non-linear changes of abundances.** (A) Predicted relative abundance dynamics of five major and minor amplicon sequence variants smoothed by the local polynomial regression method. Grey boundaries denote confidence intervals. (B) Association between amplicon efficiencies and the energy of secondary structures.

While the temporal dynamics at the genus level is monotonic, the behaviour of individual ASVs is not that straightforward (Firmicutes' composition in Fig. 2A). For example, temporal dynamics of individual relative abundances for the top five rare and abundant variants are described by nonlinear curves (Fig. 3A). More importantly, these curves intersect at some PCR cycles and diverge in others, which means the spurious similarity/difference between ASVs abundancies depending on the cycle number. This observation further stresses, that it is absolutely crucial to account for the compositional nature of HTS data.

## Investigating putative sources of amplification bias

As we have mentioned in the introduction, it is believed that differences in GC content or secondary structure of DNA templates could be responsible for the amplification bias (*Paliy & Foy, 2011*; *Kalle, Kubista & Rensing, 2014*; *Kebschull & Zador, 2015*; *Fan et al., 2019*). We also tested the assumption that sequence similarity is a good predictor of effectivity of amplification. But the Mantel test showed no significant correlation between pairwise sequence similarity (measured as Levenstein edit distance) and difference in log-transformed amplification efficiencies.

To evaluate the remaining potential predictors, we fitted a robust linear model for amplification efficiencies $\lambda$ parametrised by GC content and the energy of secondary structure $E$:

$$\log(\lambda) \sim GC + E.$$

We observed that GC content was not associated with amplification efficiencies, while the energy of secondary structure demonstrated significant association with efficiencies ($p$-value = 0.0002).

## DISCUSSION

The null hypothesis in PCR amplification is the stability of starting alleles abundances in a series of successive amplification cycles in which the absolute number of DNA copies doubles each time. If the amplification efficiencies of alleles are not the same, their abundances in the process of amplification can change in different directions. This effect, sometimes resulting in drastically distorted initial abundances, is one of the possible mechanisms of PCR bias. But the problem gets harder because of compositional effects (roots of which lay entirely in the area of statistics), which make the dynamic of allele abundances nonlinear. If in the case of PCR bias, much has been done on the issue, much less done in the area of compositional effects in metagenomics, but there is literally nothing about their interplay with PCR bias, where these effects pose almost intractable problems for the researcher. We have tried to suggest an appropriate way to deal with that in this work.

The model proposed in this work does not account for temporal changes in amplification efficiencies and, thus, might not be flexible enough to accurately reconstruct unbiased community compositions, though it would take more studies involving calibration experiments on a large set of diverse mock communities to verify these suspicions. It is perhaps more important to consider the practical side of the question. The calibration experiment we have designed and demonstrated is laborious and cost-consuming to become a widely used routine solution to the amplification bias problem. It is, of course, possible to reduce the number of replicates per cycle and, by extension, cut costs by improving innate model robustness (for example, model balances as a heavier-tailed multivariate t-distribution instead of a less robust multivariate Gaussian) and pooling PCR samples (to reduce inter-reaction variance), but these shortcuts might introduce unforeseen complications of their own. With or without any modifications, it would still take dozens of libraries and a lot of extra effort to mitigate PCR bias in a single biological sample effectively negating the two major selling points of amplicon sequencing: simplicity and accessibility. A more pragmatic route to follow might be predicting amplification efficiencies and then utilising them to estimate, in line with our model, the initial composition of the community (Expression 6).

As shown earlier, in addition to the obvious reasons for the differences in amplification efficiency (*e.g.*, GC-content, target sequence length, sequence base composition, primer sequences, and specificity, buffer compositions, presence of PCR inhibitors in the template DNA solution, cycling conditions, and thermostable DNA polymerase), there are some inconspicuous reasons, for example, the composition of the initial community (*Gonzalez et al., 2012*). It was shown that "rare biosphere" taxa tend to be lost at the final steps of PCR. The present work does not directly work with this phenomenon because the design of the experiment deals only with late PCR cycles, where the "rare taxa" probably have

already been undetectable. However, we may speculate that if the amplification efficiency of rare taxa is high enough, these taxa could still be detectable at late PCR steps.

The energy of the secondary structure was demonstrated as significantly associated with amplification efficiency, and this factor could indeed be of the primary importance. If we go through the above-mentioned list of other factors, we can find a foundation that almost all of them, one way or another, can act through the influence of secondary structures. Only the initial taxa composition, as a factor of amplification efficiency (*Gonzalez et al., 2012*), has nothing in common with secondary structure energy and may act like the positive frequency-dependent selection in evolution.

Ultimately, the data obtained inspire some optimism—both bias and compositional effects can be worked on, but the issue still requires careful study.

## CONCLUSIONS

Here we have presented a calibration experiment and a model to evaluate, characterise and mitigate amplification biases in amplicon sequencing libraries. Our model is in line with the theory of compositional data analysis, which is most suitable nowadays to describe changes in proportions of components in a microbial community. Using a human gut microbial community, we demonstrated that the model was capable of accurate capturing and reconstructing temporal dynamics of a 16S rRNA community profile undergoing multi-template "biased" PCR. We have shown that biases dramatically disturb ratios between community components, and these changes are non-linear across the cycles. The approximated unbiased community profile shows that standard library preparation workflows can alter most individual relative abundances by a factor of 2-8 and dramatically perturb high-level community composition. With the model proposed in this work, we showed the significant association of the PCR-bias with energies of secondary structures of DNA templates through the amplification efficiencies of templates. Finally, it is important to highlight that the compositional effects are universal for metagenomic studies and affect dynamics of all types, whether it be PCR amplification or natural changes.

## ACKNOWLEDGEMENTS

The author would like to thank Juan José Egozcue and Vera Pawlowsky-Glahn for CoDA expertise. Additional thank to Galah and Cockateil for helpful discussions.

### Funding

The study was funded by the Russian Science Foundation (RSF) grant number 18-16-00073-P. The funders had no role in study design, data collection and analysis, decision to publish, or preparation of the manuscript.

## Grant Disclosures

The following grant information was disclosed by the authors:
Russian Science Foundation (RSF): 18-16-00073-P.

## Competing Interests

The authors declare there are no competing interests.

## Author Contributions

- Ilia Korvigo conceived and designed the experiments, analyzed the data, prepared figures and/or tables, authored or reviewed drafts of the article, and approved the final draft.
- Anna A. Igolkina analyzed the data, prepared figures and/or tables, authored or reviewed drafts of the article, and approved the final draft.
- Arina A. Kichko performed the experiments, authored or reviewed drafts of the article, and approved the final draft.
- Tatiana Aksenova performed the experiments, authored or reviewed drafts of the article, and approved the final draft.
- Evgeny E. Andronov conceived and designed the experiments, performed the experiments, authored or reviewed drafts of the article, secured funding, and approved the final draft.

## Human Ethics

The following information was supplied relating to ethical approvals (*i.e.*, approving body and any reference numbers):

This study was approved by the Institutional Review Board of Federal State Budget Scientific lnstitution "All-Russian Research Institute of Agricultural Microbiology" (FSBSI ARRIAM) #96/06 from 04.06.2020.

## DNA Deposition

The following information was supplied regarding the deposition of DNA sequences:

Raw sequencing data are available in the NCBI Sequence Read Archive (SRA): PRJNA545409.

## Data Availability

All code and metadata required to reproduce the study are available at GitHub: https://github.com/arriam-lab2/pcr_bias_publication.

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
