# Peer review of "Be aware of the allele-specific bias and compositional effects in multi-template PCR"

_PeerJ, doi:10.7717/peerj.13888_

## Round 0.1 · original submission · Major Revisions

Dear Dr. Korvigo and colleagues:

Thanks for submitting your manuscript to PeerJ. I have now received three independent reviews of your work, and as you will see, the reviewers raised some major concerns about the research (and the manuscript). Despite this, these reviewers are optimistic about your work and the potential impact it will have on research utilizing bacterial SSU (16S) rDNA sequencing. Thus, I encourage you to revise your manuscript, accordingly, taking into account all of the concerns raised by both reviewers.

Most importantly, two of the three reviewers were more optimistic about your work, while one reviewer seems to think that the objectives of the manuscript fall short and need to be refocused. This reviewer also feels that solutions and strong conclusions are lacking. Please address these issues.

General concerns include lacking references, a lack of information with some of the figures, and a lack of clarity regarding the presentation of the experimental data (e.g., how was the data used to facilitate model evaluation and parameter estimation?).

There are many minor problems pointed out by the reviewers, and you will need to address all of these and expect a thorough review of your revised manuscript by these same reviewers.

I agree with the concerns of the reviewers, and thus feel that their suggestions should be adequately addressed before moving forward. Please note that Reviewer 1 kindly provided a marked-up version of your manuscript.

Therefore, I am recommending that you revise your manuscript, accordingly, taking into account all of the issues raised by the reviewers.

I look forward to seeing your revision, and thanks again for submitting your work to PeerJ.

Good luck with your revision,

-joe

Reviewer 1 ·

Basic reporting

This ms presents work on a problem which is the biases that PCR introduces in the detection and quantitation of amplicons for microbial community (and microbiomes) surveys and analysis of the microbial community structure.
Although PCR has many potential biases from failing primer universality, GC content, chimera generation, different target competence, etc., and references on all these problems are very limited and should be expanded greatly. Above all, a major problem of concern with this ms is the issue on biases due to different amplification efficiency on targets at different proportion in the community. This is a key-point of the ms and a previous reference demonstrating experimentally and with models this problem should be mentioned in the ms and discussed to show the advancement of the state of the art and complement results and modelling advancements. The ms I am referring is: Gonzalez et al. 2012. Amplification by PCR artificially reduces the proportion of the rare biosphere in microbial communities. Plos One 7 (1): e29973. doi: 10.1371/journal.pone.0029973. This ms showed experimentally and modeled that amplification efficiency is a function on the fraction of the representation of target sequences in the whole pool of amplifiable sequences in a sample. That study focus mainly on those low represented sequence targets (i.e., rare fractions of the microbial community) and the current ms appears to focus mainly on those highly abundant members of the community. Literature references and the background to introduce and discuss this ms should be greatly improved.
Another major problem on this ms is the lack of solutions and conclusions (see, for instance Discussion and Conclusions). This ms fails to propose solutions and represents a very preliminary study which should be continued and completed before attempting to present it.

Experimental design

The amplification efficiency during PCR depends on the fraction of each target sequence. The authors should analyze different microorganisms (sequences) present at relative abundances in their microbiomes/microbial communities. For instance, the authors should analyze those present at high abundance (i.e., >20% as an indicative), those at medium abundances (i.e., <20%) and also those at low abundances (i.e., <2% and <<<2%). In this way, the ms will profit on a complete analysis of the full range of abundances in microbial communities. As it is, the authors analyze the most abundant or major groups in their microbiomes.
The study should aim to provide solutions, methods of analysis, or strategies to solve the problem. As it is, the ms just proposes a problem (which has been proposed and demonstrated before). A more careful analysis of the problem will contribute to reach possible solutions or strategies to cope with the problematic of changing relative abundance of memebrs of the microbial communities during the amplification of their rRNA gene sequences by PCR. A redesign of the experimental problem might help to deepen in the issue and reach advanced conclusions.
It is confusing how the authors sample/measure the different target sequences during the amplification and the confusion between PCR amplification and quantitative real-time PCR amplification.

Validity of the findings

The topic of the ms is of interest but solutions or major novel contributions should be provided. There is no point to say again what has been demonstrated and fail to provide relevant contributions with solutions or strategies to solve or diminish the problematic involving in PCR.
The conclusion that more work is required and that the results do not provide with additional or significant contributions is not acceptable.

Additional comments

I would recommend to rethink the ms objectives and analyses (see above) so that they can generate a much more complete response and solution to the proposed issue during PCR amplifications on microbial community surveys.

Reviewer 2 ·

Basic reporting

This is a very well written article. All appropriate sections are included, introduction provides very good information on the past studies and describes the problem that the authors address in this work.

Figures are legible but can be improved by not using different shades of the same color hue.

Experimental design

Article describes original research, and research question is well defined in the introduction.

Methods section includes some troubleshooting steps which is great information often missing from other manuscripts. Appropriate data processing and normalization methods were used, and PCR amplification model is described with sufficient mathematical detail. Processing accounted for compositional nature of the data.

There are nevertheless some minor issues with the methods section:

Sample donor cannot be identified as one of the authors - that is against HIPAA regulations. Please include information on the institutional review board approval of the human subject research.
Please include specific primer sequences, not just reference a previous study.
How many raw reads per sample were obtained (average, min-max range)? Were the reads rarefied after filtering?
Please define parameters and coefficients in the equations shown in methods sections - some are only defined in results (e.g., "c")
Indicate that ggplot2 was used in R.
I could not locate the Bioproject in SRA archive.

Validity of the findings

Overall I found the manuscript interesting to read. The combination of experimental and modeling approaches provides powerful opportunities to infer model parameters and adjust its performance based on the actual experimental data.

However, there seems to be a significant focus on the modeling part of the study, and the sequencing information is described very briefly, and often it is difficult to understand how exactly experimental data was used to facilitate model evaluation and parameter estimation.

Why so little of sequencing results is reported? I only see a single figure 1 that shows sequencing data, but at phylum level. However, figure 3 shows prediction of the bias for many lower taxons. Why are no experimental data for these taxons presented?

How did authors define amplification efficiencies for different taxa? The model predicts changes for specific taxa, thus their amplification efficiencies had to be defined specifically for each OTU/ASV. how was this accomplished - using Bayesian inference, but on what?

In the section on the amplification bias, was this done at OTU/ASV level, or at the level of phyla? Looking at figure 2, this result suggests that many ASVs among Firmicutes were highly abundant and experienced negative bias. This seems somewhat unlikely.

I also found it very odd that there were no ASVs based on figure 4 that had little to no bias. Again, prom probabilistic theory, that is pretty unusual/unlikely. Maybe figure 4 should include all ASVs (or ASVs combined at the genus level) so that a reader can see the full picture (?)

Can authors expand in the discussion/conclusion on the possible suggestions on how to deal with the revealed biases? I agree with authors that it is not practical to run multiple PCR reactions at different # of cycles. How can one predict amplification efficiencies for any OTU/ASV?

·

Basic reporting

no comment

Experimental design

no comment

Validity of the findings

It would be key to understand what other steps you are thinking to do further with these results and the recommendations based on your results more cleared. Based on the observed that it's not recommended to do 35 cycles for example, what's the recommendation to optimize the PCR cycles to avoid these alterations. Were you able to find this optimization or not yet?

Additional comments

it's great topic that we must solve the PCR bias in 16S rRNA. The statistical modeling done has been well refuted and it's key to have this type of experiment in different times points to see if the outcome might be same results.

---

## Round 0.2 · accepted · Accept

Dear Dr. Korvigo and colleagues:

Thanks for revising your manuscript based on the concerns raised by the reviewers. I now believe that your manuscript is suitable for publication. Congratulations! I look forward to seeing this work in print, and I anticipate it being an important resource for groups studying the use of bacterial SSU (16S) rDNA sequencing. Thanks again for choosing PeerJ to publish such important work.

Best,

-joe

Reviewer 1 ·

Basic reporting

The ms have improved the ms following the Reviewers' comments.
Some minor corrections should be considered:
- References should be ordered alphabetically.
- Gonzalez is spelled ending in s and in z.
- Some additional formatting should be revised.

Experimental design

Ok

Validity of the findings

Ok. In the future, the authors might want to continue the study of energy of secondary studies and amplification efficiency relationship.

Additional comments

Ok